



# Measuring the spatiotemporal variability of snow depth in subarctic environments using unmanned aircraft systems (UAS) – Part 2: Snow processes and snow-canopy interactions

Leo-Juhani Meriö[1], Anssi Rauhala[2], Pertti Ala-aho[1], Anton Kuzmin[3], Pasi Korpelainen[3], Timo Kumpula[3], Bjørn Kløve[1], Hannu Marttila[1]

[1]Water, Energy and Environmental Engineering, Faculty of Technology, University of Oulu, 90014, Finland
[2]Civil Engineering, Faculty of Technology, University of Oulu, Finland.
[3]Department of Geographical and Historical Studies, University of Eastern Finland, Joensuu, Finland.

*Correspondence to*: Leo-Juhani Meriö (leo-juhani.merio@oulu.fi)

**Abstract.** Detailed information on seasonal snow cover and depth is essential to the understanding of snow processes, operational forecasting, and as input for hydrological models. Recent advances in unmanned aircraft systems (UASs) and structure from motion (SfM) techniques have enabled low-cost monitoring of spatial snow depth distribution in resolutions up to a few centimeters. Here, we study the spatiotemporal variability of snow depth and interactions between snow and vegetation in different subarctic landscapes consisting of a mosaic of conifer forest, mixed forest, transitional woodland/shrub, and peatland areas. To determine the spatiotemporal variability of snow depth, we used high-resolution (50 cm) snow depth maps generated from repeated UAS-SfM surveys in the winter of 2018/2019 and a snow-free bare ground survey after snowmelt. Due to poor sub-canopy penetration with the UAS-SfM method, tree masks were utilized to remove canopy areas and the area (36 cm) immediately next to the canopy before analysis. Snow depth maps were compared to the in-situ snow course and a single-point continuous ultrasonic snow depth measurement. Based on the results, the difference between the UAS-SfM survey median snow depth and single-point measurement increased for all land cover types during the snow season, from +5 cm at the beginning of the accumulation to -16 cm in coniferous forests and -32 cm in peatland during the melt period. This highlights the poor representation of point measurements even on the sub-catchment scale. The high-resolution snow depth maps agreed well with the snow course measurement, but the spatial extent and resolution of maps were substantially higher. The snow depth variability (5–95 percentiles) within different land cover types increased from 17 cm to 42 cm in peatlands and from 33 cm to 49 cm in the coniferous forest from the beginning of the snow accumulation to the melt period. Both the median snow depth and its variability were found to increase with canopy density; this increase was greatest in the conifer forest area, followed by mixed forest, transitional woodland/shrub, and open peatlands. Using the high spatial resolution data, we found a systematic increase (2–20 cm), then a decline of snow depth near the canopy with increasing distance (from 1 m to 2.5 m) of the peak value through the snow season. This study highlights the potential of the UAS-SfM in high-resolution monitoring of snow depth in multiple land cover types and snow-vegetation interactions in subarctic and remote areas where field data is not available.



## 1 Introduction

Snow cover is of great importance for northern ecosystems and hydrology, providing shelter for plants and animals in the harsh winter conditions and maintaining freshwater resources and seasonal hydrological processes (Pomeroy and Brun, 2001; Mankin et al., 2015; Blume-Werry et al., 2016). The water that is stored in the snowpack during the winter is released in spring freshet, recharging groundwater, soil water, and ultimately maintaining low flow conditions in early summer (Earman et al., 2006; Godsey et al., 2014; Meriö et al., 2018). Additionally, snow conditions are essential for several ecosystem services, including recreational and tourism uses (Scott et al., 2008; Neuvonen et al., 2015). Numerous studies have documented the changes and their regional variability in snow conditions in Finland (Luomaranta et al., 2019) and the Northern Hemisphere (Brown and Robinson 2011; Pulliainen et al., 2020). The ongoing environmental change and future projections involve snow conditions that are changing rapidly in high latitude regions (Musselman et al., 2017; Mudryk et al., 2020). Snow processes are known to have high spatiotemporal variability, thus more detailed high-resolution knowledge of snow accumulation and melt is needed to support process understanding, modeling, forecasting, and decision making.

The spatiotemporal variability of snow accumulation is governed by climate, terrain characteristics, and vegetation cover in scales above 100 m and by wind redistribution, microtopography, and canopy interception in scales below 100 m (McKay and Gray, 2004). Forested and open areas have different snow accumulation and melt characteristics (Pomeroy et al., 2002; Gelfan et al., 2004). In forests, the snow depth has been found to depend on forest cover with less snow in the denser forest because of canopy interception and sublimation (Varhola et al., 2010), and in forest openings on their size, with most snow accumulating in clearings 2–5 times the height of nearby trees (Pomeroy et al., 2002). In larger open areas, wind erosion and drift redistribute the snow on the sheltered forest edges where the wind speeds are reduced (Hiemstra et al., 2002). Snowmelt is governed by the energy available for melt, mainly influenced by topography and vegetation (Jost et al., 2007). Generally, snowmelt rates are lower in shadowed areas, like topographic depressions, northern slopes, and areas shaded by dense forest canopies (Gary, 1974; Clark et al., 2011). However, longwave radiation from forest canopies can also increase the melting speed near the trees (Golding and Swanson, 1978). Additionally, the timing of snow depletion is dependent on the amount of pre-melt snow (Liston, 1999; Faria et al., 2000).

Currently, there are various techniques to monitor snow properties, such as snow cover, depth, and snow water equivalent (Kinar and Pomeroy, 2015). Snow courses and measurement networks are established to improve the poor representation given by single-point measurements, but small-scale or even regional spatial variability is not captured using these techniques. Satellite remote sensing products extend the scale of the measurements to a large scale (Dietz et al., 2012), but the resolution of the mature products is coarse (~25 km). Methods for higher resolution satellite products are continuously under development. For example, Lievens et al., (2019) showcased a method for 1 km resolution snow depth retrieval for mountain regions. Airborne LiDAR from manned aircraft provides high-resolution snow depth for relatively large areas but with high cost (Deems and Painter, 2006; Deems et al., 2013). The recently popularized Unmanned Aerial Systems, together with Structure from Motion (UAS-SfM) techniques, have shown the potential for cost-efficient solutions for high-resolution snow




depth mapping (see accompanying article Part 1, Rauhala et al., 2022, submitted to the same journal; Vander Jagt et al., 2015) enabling new methods of snow process research.

Previous UAS-SfM studies have focused on testing the accuracy of the method on mountainous regions or snow on tundra, glaciers, prairies, and meadows, with mostly low (grassland, shrub, bushes) or non-vegetated surfaces (see Rauhala et al., 2022, submitted). Snow process observations in UAS-SfM studies have mostly been related to topographic features such as

aspect, cornices, gullies, exposed ridgelines, and broad elevated slopes (Bühler et al., 2016; Redpath et al., 2018; Niedzielski et al., 2019) and wind redistribution, snow erosion and tree wells, which are the spaces under spruce trees which receive less snow than their surroundings (Harder et al., 2020). Lendzioch et al., (2016) evaluated snow depth in an open and small forested area in Sumava National Park, Czech Republic, and found the accuracy was better in open areas than forested areas due to deadwood on the ground and vegetation effects. Niedzielski et al. (2019) observed snow depths in sites covering forests,

meadows, and arable land in Poland, but these forested areas were removed as outliers from the snow depth maps. Sub-canopy penetration of UAS-SfM was compared to UAS-LiDAR in subalpine areas and prairies in Canada with the conclusion that UAS-SfM is not capable of observing snow depth below the canopy (Harder et al. 2020). Most recently, Schirmer and Pomeroy (2020) studied the association between snow depth differences during the ablation period and snow cover brightness, slope, and initial snow depth at Alpine ridge in the Canadian Rocky Mountains. To our knowledge, no studies have used the high-

resolution data from UAS-SfM to study snow accumulation and ablation processes, and interactions between vegetation and snow in the subarctic boreal region.

The overall aim of this study was to evaluate the variability of snow accumulation and melt in high spatial resolution using UAS-SfM. With these novel datasets, we studied interactions between snow cover and vegetation in different subarctic land cover types. The specific research questions were: 1) Can UAS-SfM map the spatial snow depth variability at a high resolution

throughout the snow season in demanding weather and light conditions in the subarctic environment, 2) how do snow accumulation, redistribution, and melt differ in forested and open mire landscapes, and 3) what interactions between snow and vegetation can be revealed with high spatial resolution UAS-SfM snow depth surveys.

## 2 Study area

Three test sites, mire (14.41 ha, Fig. 1c), mixed (15.40 ha, Fig. 1d) and forest (15.87 ha, Fig. 1e), with varying landcover were

selected at a snow course transect in Lompolonjänkä catchment (68.00° N, 24.21° E) (Marttila et al., 2021), adjacent to Pallas-Yllästunturi National Park in the subarctic region (Fig. 1). The land cover in the catchment consists mostly of boreal coniferous forests and open peatlands. The most common tree species are Norway spruce (Picea abies (L.) H. Karst) with occasional Scots pine (Pinus sylvestris L.), downy birch (Betula pubescens Ehrh.), and mountain birch (Betula pubescens ssp. czerepanovii) (Sutinen et al., 2012). On open peatlands, as well as at times in the forested areas, there are occasional bushes and other low

vegetation. The mire site consists mostly of flat open peatland, the forest site of gently sloping coniferous forest and the mixed site of a relatively flat mixture of both. Elevation in the study area varies from 267 to 350 masl, the slope varies between 0–



4.76 degrees and the aspect is towards the west-northwest. Peatland areas, at the mire and mixed sites, are almost flat with a slope of 0–0.25 degrees, while the slope is highest in parts of the forested area at the forest site.

Fig. 1. (a) The location of the study area south of Lake Pallasjärvi and east of the Lommoltunturi fells. The location of Fig. 1 (b) is highlighted by the white rectangle. Hillshade courtesy of National Land Survey of Finland. (b) Locations of the manual snowline measurement, ultrasonic point sensor, and outlines of the subplots (sites mire, mixed and forest read from northwest to southeast) within the catchment. Figs (c), (d), and (e) zoom in to the mire, mixed, and forest subplots, respectively. Orthophoto courtesy of National Land Survey of Finland.





Typically, stable snow cover in the area during 2006–2018 (the period of record) has started building in mid-October, with peak accumulation (96 cm) at the beginning of April just before the melt season, and all snow having melted by the end of May (Fig. 2). During the study period, stable snow cover was established in late November and the snow depth stayed significantly under its mean until the peak accumulation (100 cm on 23 March), when it slightly surpassed the long-term mean

value. The melting period started rapidly under due to a warm spell in early April, all snow had melted a few days earlier (26 May) than the average in the period of record 2006–2018. The mean annual temperature for the hydrological year (Oct– Sep) 2019 was 0.5 °C (0.4 °C in 2004–2018). Precipitation was 621 mm (638 mm in 2008–2018), which was made up of 40 % snowfall (42 % in 2008–2018) (1.1 °C used as the threshold for snowfall (Feiccabrino & Lundberg, 2008; Jenicek et al., 2016)). Open weather data from the Finnish Meteorological Institute (FMI) Kenttärova measurement station was used for climate

parameter calculations.

**Fig. 2. Typical snow conditions in the study area (Finnish Meteorological Institute (FMI) Kenttärova point measurement) between 2006-2018. and during the study winter of 2018–2019 (in red). Survey times are shown with vertical lines (dark green). Survey data**
**from June 2018 was used to determine the ground digital elevation model.**





## 3 Materials and methods

### 3.1 UAS campaigns and reference measurements

Data from five of the seven total UAS campaigns were selected for snow process analysis. Two of the surveys were discarded due to challenges during the campaigns that hindered the data collection from all study plots. During the snow period, the remaining surveys were conducted in varying snow, weather, and light conditions. They were held at the beginning (10–13 December 2018, DEC-12) and middle (18–22 February 2019, FEB-21) of the snow accumulation period, one at the beginning of the melt period (1–5 April 2019, APR-03, shortly after peak accumulation) and one in the middle of the melt period (22–25 April 2019, APR-24). The last survey (04–05 June 2019), for the snow-free conditions, was conducted before ground vegetation growth (when the vegetation was still compressed) approximately one week after all snow had melted.

The aerial surveys were done using four drones: DJI Phantom 4 RTK (real-time kinematic) quadcopter (P4RTK), DJI Mavic Pro, DJI Phantom 4, and eBee Plus RTK. We selected data from P4RTK because it provided the highest accuracy for snow and ground surface maps created using the SfM photogrammetry technique (see Rauhala et al., 2022, submitted). The flight height target was 110 m, which provided ~3 cm ground resolution. Forward and side overlap targets were at least 80 % and 75 %, respectively, for aerial pictures.

Before starting the aerial surveys, an average of 13 ground control points (GCP) (8–17, median 14) and 16 random checkpoints (CP) (6–38, median 15) were marked and measured using RTK GNN receivers (Trimble R10 and Topcon Hiper V) at each test plot during all campaigns. For the RTK equipped drones, we selected the GCP that was nearest to the flight control for the snow/ground surface map calculations. All GCPs were needed when using data from non-RTK drones. CPs were used to estimate the accuracy of the snow and ground surface models.

During every UAS survey, reference snow depth was manually measured from a snow course (46 stationary points with a mean distance of 50 m between points) transecting the study plots, with an accuracy of +/- 2 cm. In addition, data from an automatic ultrasonic snow depth sensor (Campbell Scientific SR50-45H) with an accuracy of +/- 1 cm, located in the forest plot at the highest elevation of the study area and operated by FMI, was used as a reference to compare the UAS-derived snow maps. More detailed information for the UAS campaigns and flight parameter selection can be found in the accompanying article (Part 1) (Rauhala et al. 2022, submitted).

### 3.2 High-resolution snow depth maps and tree mask

The principal technique for snow depth map generation was subtracting snow surface elevations from snow-free ground elevations. Agisoft Photoscan/Metashape Professional v.1.4.5/v1.6.0 software (Agisoft, 2019) utilizing an SfM-technique was used to create surface elevation maps using high-quality and moderate depth filtering settings. This resulted in full resolution (~3 cm) orthomosaic and 2 x full-resolution snow and ground surface maps or digital surface models (DSMs). The processed data was exported as georeferenced files to ArcGIS 10.6 (Esri, 2019) for further processing.





Due to poor sub-canopy penetration when using the UAS-SfM method (Harder et al., 2020), we omitted data at tree locations, and immediately next to trees using special tree masks. Tree masks were generated using Maximum Likelihood Supervised Classification in ArcGIS 10.6 and full resolution orthomosaics from the survey conducted on 3 April 2019. We selected this
survey because snow had melted from the tree canopies, giving a clear contrast between trees and snow. This was then used for classifying the data. SfM method had challenges with differentiating trees from snow cover from the data for surveys in which the canopies were covered by snow. This led to artificially increased snow depths next to the tree branches. Moreover, the deciduous trees without leaves were problematic in supervised classification because bare branches were easily mixed with shadowed snow cover, leading to the classification of shadowed snow cover as canopy or canopy as snow cover. To mitigate
these methodical challenges, we tested different buffer distances around the classified tree mask and found that 36 cm was a good trade-off for removing the compromised zones next to trees without losing too much valuable snow cover data. After buffering, the tree masks were saved with a resolution of 2 cm and applied to snow and ground surface maps before snow depth calculation.

Snow depths were calculated for each pixel by subtracting bare ground (snow-free) elevation from snow surface elevation for
each survey carried out in the snowy season, resulting in DEM of differences (DoDs). Snow depth maps were aggregated to a 50 cm resolution before further data analysis. This resolution was chosen to smooth small-scale variability while keeping a reasonably high resolution for snow-vegetation interaction analysis. Moreover, the selected resolution followed findings from De Michele et al., (2016) where the standard deviation of snow depth increased with a decreased pixel size but stabilized for resolutions smaller than 1 m. For analyzing snow depth variability compared to point measurement, anomaly maps were
created by subtracting the corresponding snow depth measured with ultrasonic sensors from each pixel of the UAS-SfM derived snow depth maps. The snow depth calculated using ultrasonic sensors was also subtracted from snow course measurements. The full workflow for tree mask and snow depth map generation, along with their calculated accuracy, is presented by Rauhala et al. (2022, submitted).

### 3.3 Land cover and snow processes

Corine land cover 2018 data with a resolution of 20 m (SYKE, 2019) was further used to study the snow processes for different landcover types: coniferous forest (10.07 ha), mixed forest (1.42 ha, made up of mixed (1.34 ha) and broadleaved forest (0.08 ha) land cover types), transitional woodland/shrub (1.06 ha) and peatbogs (10.37 ha) (Fig. 3). The Corine land cover dataset was selected for this purpose due to its i) approved vegetation classification and ii) availability across the Eurasian region. To further study the interactions between canopy cover and snow depth, Euclidian distance from the nearest tree mask pixel,
representing the canopy, was calculated for each snow pixel in ArcMap 10.6 (Fig. 3). These distance masks were used to calculate median snow depth as a function of distance from the canopy for each land cover type.

Histograms, boxplots, statistical indicators (median, mean, and 5–95 percentiles), and tests were used to study snow depth variability and differences within and between different land cover types. The Kruskal–Wallis (K–W) test was selected to find whether there was a difference between group (land cover types) medians. To find which groups might differ, a Dunn's test





(Dunn, 1964) with a Bonferroni adjustment (Dunn, 1961) was used. The large sample size (in total 917,045 snow pixels for each survey) caused problems in statistical analysis, as large sample sizes may be too big to fail (Lin et al., 2013). To mitigate the problem, we additionally ran the Monte-Carlo simulation by extracting 100 samples of sizes ranging from 100 to 4000, at an increment of 100, from each land cover type data set and ran the K-W and Dunn's test for each sample to generate coefficient/p-value/sample-size (CPS) charts.




**Fig. 3. Corine land cover (2018) data (above) and distance from tree mask (below) (calculated using a Euclidian distance tool in ArcMap 10.6) for test sites. "Commercial units" refers to measurement infrastructure in the peatbog.**

## 4 Results

### 4.1 Spatiotemporal variability of snow depth during accumulation and melt

The snow depth anomaly in test sites compared to ultrasonic point measurements exposed high spatial variability in snow depth and differences to the point measurement (Fig. 4, Table 1). At the beginning of the snow accumulation (DEC-12), this





difference was positive, showing slightly more snow (median +0.05 m) in all sites compared to the point measurement reference. In the middle of the snow accumulation season (FEB-21), however, the difference was slightly negative (median -0.03 m). The negative difference increased (median -0.09 m) at the beginning of melt (APR-03), reaching a peak during the middle of the melt period (APR-24), where a high negative difference (median -0.22 m) was observed. This difference was highest in the mire site (median -0.28 m) but was also present in the mixed site (median -0.24 m), a partly covered open peatbog similar to the mire site, where the forest cover was more variable, including increased mixed forest and transitional woodland/scrub areas. In open peatlands, the snow had accumulated on the forested edges of the open areas. During the melt (APR-24), the forest site showed the smallest difference (median -0.08 m) compared to the point measurement. The reference snow course measurements had similar snow depth central values and distribution compared to the UAS-SfM derived snow depths (Fig. 5 and Tables 1, 2, and 3).





**Fig. 4. Snow depth (P4RTK snow - P4RTK ground) difference compared to point measurement at forest site. The point measurement** 210 **is marked with black dot in the forest site in the upper-right map. Snow depth from the point sensor is shown in the y-axis, after the survey date. Cold colors indicate areas where the snow depth is lower than the point measurement.**





**Table 1. Median differences of snow depth (cm) between point measurement and UAS-SfM derived DoDs for each test site and snow course measurements.**

| Date | Mire | Mixed | Forest | All | Snow course |
|------|------|-------|--------|-----|-------------|
| Dec-12 | 6 | 4 | 5 | 5 | 1 |
| Feb-21 | -4 | -4 | 4 | -3 | -4 |
| Apr-03 | -11 | -9 | -4 | -9 | -3 |
| Apr-24 | -28 | -24 | -8 | -22 | -21 |

### 4.2 Land cover effect on snow depth variability

UAS-SfM derived snow depths on DEC-12 were slightly higher compared to point measurements (median 0.05–0.06 m) and similar for all Corine landcover types in the test area. The highest variability was found in the mixed and coniferous forest (5–95 % range 0.35 m and 0.33 m, respectively), while in open peatlands and transitional woodland/shrub the variability was lowest (5–95 % range 0.17 m and 0.19 m, respectively) (Fig. 5 and Tables 2 and 3). In the middle of the accumulation period, FEB-21, the snow depth was lower in open peatlands and transitional woodland/scrub (median -0.06…-0.02 m) than mixed and coniferous forests (median 0.01 m) with high variability in forested areas (0.62–0.66 m). For the beginning of the snowmelt period (APR-03), the median snow depths were lower compared to point measurements for all land cover types, with the highest negative difference in open peatlands (median -0.13 m); for the other areas this difference was smaller (-0.06…-0.05 m). The spread of the snow depth was similar (0.32–0.35 m) for all land covers. The biggest difference and variability between landcover types was observed in the middle of the melt period (APR-24). Again, the difference was highest in open peatland (median -0.32 m) followed by transitional woodland/shrub (-0.22 m). In mixed and coniferous forests, the difference was lower (median -0.16 and -0.13 m, respectively). The variability of the snow depth was high for all landcover types (0.42–0.49 m). The snow depth variability increased throughout the snow season, except for in mixed and coniferous forest areas in FEB-21, when the variability was at its highest of all the surveys (Table 3).

For the full data set, the Kruskal–Wallis test showed significant ($p < 0.001$) differences between land cover types for median snow depth in all surveys from DEC-12 to APR-24, with increasing chi-squared values, 3391, 86489, 92497, and 237345, respectively. Dunn's post hoc test with a Bonferroni adjustment showed that all median snow depths between land cover types were different from each other for all surveys. The number of observations in the full data set was 403443, 56426, 42327, and 414849 for the coniferous forest, mixed forest, transitional woodland/shrub, and peatbogs, respectively.

The CPS charts (see Figs. S1–S4 in the Supplement) show that with smaller random sample sizes the differences between snow depth medians are not that evident. For DEC-12, the CPS chart indicates how median snow depths are similar between all land cover types when the sample size is 100. With an increasing sample size, the similarity is still clearly visible for peatbog and conifer forest, and transitional woodland/shrub and mixed forest. On FEB-21, the differences in median snow depth increased, but even with a small sample size, the similarity is visible for conifer forest and transitional woodland/shrub,





and transitional woodland/shrub and mixed forest. It remained similar for conifer forest and mixed forest with a larger sample
sizes. For APR-03, peatbog land cover shows no similarity with other land cover types, but other land cover types show
similarities with each other. On APR-24 the differences between snow depth medians are highest and similarity is indicated
only for conifer forest and mixed forest with smaller sample sizes.

**Table 2. Median and mean differences of snow depth (cm) between point measurement and UAS-SfM derived DoDs for each land cover type and snow course measurements.**

| Date | Peatbogs | | Transitional wood/shrub | | Mixed (and broadleaved) | | Coniferous | | Snow course | |
|------|-----|------|-----|------|-----|------|-----|------|-----|------|
|      | Med | Mean | Med | Mean | Med | Mean | Med | Mean | Med | Mean |
| Dec-12 | 5 | 5 | 6 | 7 | 6 | 14 | 5 | 7 | 1 | 2 |
| Feb-21 | -6 | -6 | -2 | -1 | 1 | 6 | 1 | 4 | -4 | -3 |
| Apr-03 | -13 | -13 | -5 | -5 | -5 | -8 | -6 | -9 | -3 | -3 |
| Apr-24 | -32 | -31 | -22 | -2 | -16 | -18 | -13 | -16 | -21 | -29 |

**Table 3. 5% and 95% percentiles of differences in snow depth (cm) between point measurement and UAS-SfM derived DoDs and snow course measurements.**

| Date | Peatbogs | | | Transitional wood/shrub | | | Mixed (and broadleaved) | | | Coniferous | | | Snow course | | |
|------|-----|-----|---|-----|-----|---|-----|-----|---|-----|-----|---|-----|-----|---|
|      | 5% | 95% | Δ | 5% | 95% | Δ | 5% | 95% | Δ | 5% | 95% | Δ | 5% | 95% | Δ |
| Dec-12 | -4 | 13 | 17 | -3 | 16 | 19 | -8 | 27 | 35 | -11 | 22 | 33 | -4 | 11 | 15 |
| Feb-21 | -21 | 8 | 29 | -16 | 16 | 32 | -20 | 46 | 66 | -25 | 37 | 62 | -9 | 7 | 16 |
| Apr-03 | -28 | 4 | 32 | -21 | 11 | 32 | -24 | 9 | 33 | -28 | 7 | 35 | -17 | 9 | 26 |
| Apr-24 | -52 | -10 | 42 | -43 | -1 | 42 | -41 | 5 | 46 | -44 | 5 | 49 | -64 | -5 | 59 |






**Fig. 5. Snow depth difference (DoD P4RTK– ultrasonic point measurement) histograms for different Corine land cover types for all surveys. The boxplot shows difference data where Kenttärova Ultrasonic snow depth is subtracted from manual snow course measurement data in 46 locations. The dotted lines mark the median snow depths for each land cover type. Ultrasonic measurement is located at 0.0 on the x-axis.**

## 4.3 Vegetation interaction with snow depth

Median snow depth was observed to increase with distance from the canopy at a proximity of 0.5 to 3 meters (Fig. 6). This increase was moderate (3–5 cm) during the snow accumulation season (DEC-12 and FEB-21) and was reinforced during the last two surveys at the beginning (APR-03) and in the middle (APR-24) of the melt period (up to +15 cm in forested areas).





However, the increase remained moderate (2–8 cm) in peatland and transitional woodland/shrubland covers for all survey times. After the peak, the snow depth started to decrease with distance from the canopy.

In peatland land cover, the snow depth started to decrease after its maximum value. This decrease continued to a 5–10 meter distance from the canopy (Fig. 7). Subsequently, the variability of the snow depth was highest after 30 meters from the canopy, where there could be bushes, and the number of pixels decreased. For the conifer forest, there is an anomaly and highly variable

point cloud at 8–14 m from the canopy. In conifer, mixed forest, and transitional woodland/shrub the snow depth had very low values between 5 and 10 meters from the canopy.

The confidence interval (95%) of the median snow depth increases with the distance from the canopy (Fig. 8) as the number of pixels decreases. For coniferous forest and mixed forest, the confidence interval widened substantially after 4–5 m, for transitional woodland/shrub this happened after 4 m, and for peatbogs after 8 m distance from the canopy.



**Fig. 6. Zoomed Euclidian distance from 0 to 5 m from the canopy (tree mask) for Corine landcover types in the test sites. The Y-scale width is 0.26 m for all sub-figures for comparable variability, but the min and max variation is similar for DEC-12 and APR-24, and FEB-21 and APR-03.**



**Fig. 7. Median snow depth as a function of Euclidian distance from the canopy (tree mask) for different Corine land cover types in the test sites. Boxplots show the manual snow course measurement data and the dotted red line shows the measurement data from the ultrasonic sensor at Kenttärova. Histograms show the count of snow depth pixels for each land cover type as a function of distance from the tree mask.**

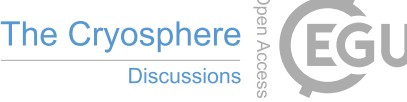



**Fig. 8. The confidence interval of median snow depth vs. distance from the canopy (tree mask) for different land cover types in survey APR-24. The 95% confidence interval for the median is marked as a light violet band in the figure. The confidence band ends at the distance where the number of observations reduces to 1 or to a very small number of observations that are similar to each other. Dark grey shows the number (n) of pixels for each distance. The dashed horizontal line marks 100 pixels.**



## 5 Discussion

### 5.1 Spatiotemporal variability of snow depth during accumulation and melt

In this study, we have successfully created high-resolution snow depth maps for the whole snowy season, covering snow accumulation and melt periods (from December to April) using the UAS-SfM method. As the generated snow depth maps were compared favorably to conventional snow course surveys, the high-resolution method extends areal coverage, providing more detailed information of snow depth distribution and data for the snow process analysis. Moreover, the spatial variability

shows the errors that may be associated with using point measurement as a regional reference for snow depth (Figs. 4 and 5). Regional snow depth/presence would be greatly overestimated during the melt season if point measurement in the forest was used for regional reference. This could have major ramifications on operational flood estimates and simulations.

The UAS-SfM studies have so far mostly focused on the accuracy and precision analysis of the method, leaving spatial snow process considerations aside. This is especially true in boreal and subarctic landscapes, which are often comprised of a mosaic

of forested, transitional, and peatland vegetated areas. However, attempts have been made to study the accuracy of the method in a defoliated spruce forest (Lendzioch et al., 2016), the accuracy of differential snow depth maps on ~50 m transects in sparse regenerating temperate broadleaved/mixed forest (Fernandes et al., 2018), and the ability of UAS-SfM to observe sub-canopy snow depth in temperate conifer forests (Harder et al., 2020). Instead of removing the forested areas or filtering the outliers from surface or snow depth maps, we used a tree mask to remove the noisy under- and near-canopy snow cover. These areas

are problematic to UAS-SfM due to the mix of pixels on the snow surface and on the canopy, the difference of the canopy diameter with and without snow-load, and broad-leaved trees that drop leaves in winter. Using the tree mask, we were successfully able to study snow depth dynamics in subarctic spruce forest areas. UAS-SfM techniques have typically been applied in single campaigns around the time of the deepest snow cover, or the focus has been on melt season. We, however, did measurements throughout the snow season, from early accumulation to melt season, allowing us to study spatio-temporal

variations in snow depth. This allowed us to quantify and compare differences in snow depth patterns and snow-canopy interaction in high resolution and different snow conditions.

### 5.2 Snow depth variability for different land cover types

We observed differences in median snow depth (+5 cm to -32 cm compared to point measurement) and its variability (from 17 cm to 49 cm) for different land cover types and found that it generally increased as the snow season progressed (Fig. 5,

Tables 1 and 2). The variability of snow depth was higher in forested areas compared to peatlands, matching the findings of Jost et al., (2007) for forests and clear-cuts. In the early phase of accumulation (DEC-12), the median snow depths were similar between land cover types, reflecting the similarity in surface texture and how the snow was trapped by the shrub on the peatlands. The low vegetation in open areas can hinder wind transport close to the ground (Liston et al., 2002), but its effect will decrease after the snow depth is above it and after that, the fallen snow is more subject to wind redistribution. However,

compared to manual snow course and ultrasonic point reference measurements, the snow depth was overestimated by an



average of +5 cm in the early phase of the snow accumulation (DEC-12). This is likely due to the snow-covered low vegetation being misclassified as snow surface (as observed in Fernandes et al., 2018), which was also visible in our orthophotos from the test sites (not shown).

In the middle (FEB-21) and the end (APR-03) of the snow accumulation season, the snow depth variability increased, and lower depths were observed, especially in peatlands (compared to other land cover types and ultrasonic point references). This can be explained by the wind transport snow process both redepositing and sublimating snow (Pomeroy et al., 2002). Similar differences between open peatlands and forested areas were observed by Meriö et al., (2018). We observed increased snow depths at the edges of the peatlands (Fig. 4, see campaigns FEB-21 and APR-03 on mire), where forested areas slow the wind speeds and the edges in proximity to the forest may act as a sink for the wind transported snow (Hiemstra et al., 2002). These

findings agreed with other studies (Hiemstra et al., 2002; Ketcheson et al., 2012).

The exceptionally high snow depth variability in conifer and mixed forests on FEB-21 (Table 3, Fig. 5), was likely the result of snow on tree canopies causing anomalies in snow DEMs near the canopy. The SfM method faced challenges in these conditions, affecting snow depths beyond our tree masks, especially near broad-leaved trees where leafless branches could only partly be identified using supervised classification and thus not removed completely by the tree mask (Fig. 4, more in

Rauhala et al., 2022, submitted).

The variability of snow depth was highest in the middle of the melt period (APR-24) between and within the land cover types, confirming earlier findings that spatial snow depth variability increases with time and scale (Neumann et al., 2006; Lopez-Moreno et al., 2015). In peatlands, the snow depth was lowest, explained by the lower initial snow depth at the beginning of the melt, likely caused by the wind drift, and the higher availability of energy for melt due to direct exposure to sunlight. The

second lowest snow depth was found in transitional woodland/shrub, also hypothesized to be caused by melt due to high solar exposure (Hardy et al., 1997). The highest snow depths were found in conifer forests followed by mixed forests. This high depth was thought to be due to open areas, less direct shortwave radiation energy, and higher initial snow depth before melt (Lundqvist and Lott 2008). In the forested areas, the canopy cover was fairly low and interception was minor during the study winter, explaining the higher snow depths.

At the end of the snow accumulation period and especially the middle of the melt period, the snow depths were substantially lower than the ultrasonic point measurement for forested and transitional land covers, highlighting the poor representativity of point measurements for similar land cover types (Fig. 5). The central value and variability of the snow depth agreed generally well with manual snow course measurements, but the UAS-SfM snow depth maps expanded the spatial coverage substantially. Nonetheless, our analysis suggests that snow course, a widely used operational method for characterizing bulk snowpack

(Pirazzini et al., 2018), produces a realistic picture of areal snow depth and its variability.

**5.3 Vegetation interaction with snow depth**

We found a systematic increase (from +2 to +15 cm) then a decline of the median snow depth near the canopy (after the 0.36 m buffer from the canopy edge). Furthermore, we found an increasing distance of the peak value through the snow season





(Fig. 6). This canopy interaction with snow cover is also documented by Pomeroy and Goodison (1997), who show how snow
depth increases 10 cm from the edge of the branches to a 2 m distance for a white spruce, in a stand of trembling aspen. Similar
findings were seen in sub-alpine forests by Musselman et al., (2008), who used normalized snow depths around trees with a
canopy radius less than 2 m and 4 m. Similar behavior is also indicated in the recent study near larch trees in Kananaskis, AB,
Canada by Harder et al. (2020). In forested areas, canopy interception and sublimation hinder the accumulation of snow under
the canopy, which also affects the fringe area. Forest openings with dimensions from 2–5 times the height of the surrounding
forest tend to collect the snow (Pomeroy et al., 2002). Tree trunks and canopies form shadowed areas but also absorb solar
radiation and emit longwave radiation that can speed up the melt near trees (Faria et al., 2000; Lundqvist et al., 2013).

Interestingly, we found that the median snow depth had a peak value around 1 m from the tree mask during accumulation
season, but this peak distance increased up to 2.5 m in the middle of the melt period. After the peak, the snow depth decreased
(Fig. 6). Moreover, the peak (from +2 cm in DEC-21 to +20 cm in APR-24) was intensified at the end of the melt period for
conifer and mixed forests. This peak was less dramatic for mires and was not observed in transitional woodland/shrub. To our
knowledge, this temporally changing canopy-snow interaction is not documented elsewhere.

For open peatland landcover, this peak may be explained by the wind distribution process that transports the snow to the edges
of the open areas, where it's trapped by trees. A slight peak was observed for transitional woodland/shrub only after snowfall
events in DEC-12 and FEB-21, but not for APR-03, when the compaction of snow after the last snowfall had occurred on or
before APR-24 when the snow was melting. This might be explained by the limited, compared with more densely forested
conifer and mixed land cover, canopy effect, which may still hinder wind redistribution compared to open areas. For conifer
and mixed forest, the peak was clear during snow accumulation, especially on FEB-21, but decreased or was non-existent at
the end of the accumulation season, APR-03 (Fig. 6). The effect was intensified in the middle of the melt period (APR-24),
assumed to be caused by the combined impact of the trunk/canopy effect (longwave melt energy) extending further from the
canopy and direct solar radiation affecting the northern side of the forest openings, while the southern sides were protected by
shadows (Faria et al., 2000; Essery et al., 2008). This may create asymmetric snow depth patterns (Fig. 4, APR-24) that are
shown as decreased snow depths after 3 m from the tree mask. Moreover, the uncertainty is increased after a 2.5–3 m distance
from the tree mask, especially for forested/transitional woodland areas because the number of pixels for those distances
decreased substantially (Fig. 8). Thus, the snow depth decrease on APR-24 after 3 m from the tree mask in forested areas
remains unexplained. The observed snow processes for different land cover types and snow-canopy interactions for snow
accumulation and melt period are summarized in Fig. 9.

The anomalies between 5 and 14 m from the canopy (Fig. 7) were thought to be partly misclassified Corine landcover pixels:
conifer forest pixels at the edge of the peatland (mire site), conifer forest pixels at the edge of the peatland (mixed site), and
the road to Kenttärova FMI measurement station (forest site), and mixed forest pixels at the edge of the peatland (mire site)
and transitional woodland (shrub pixel at edge of the peatland (mixed site).



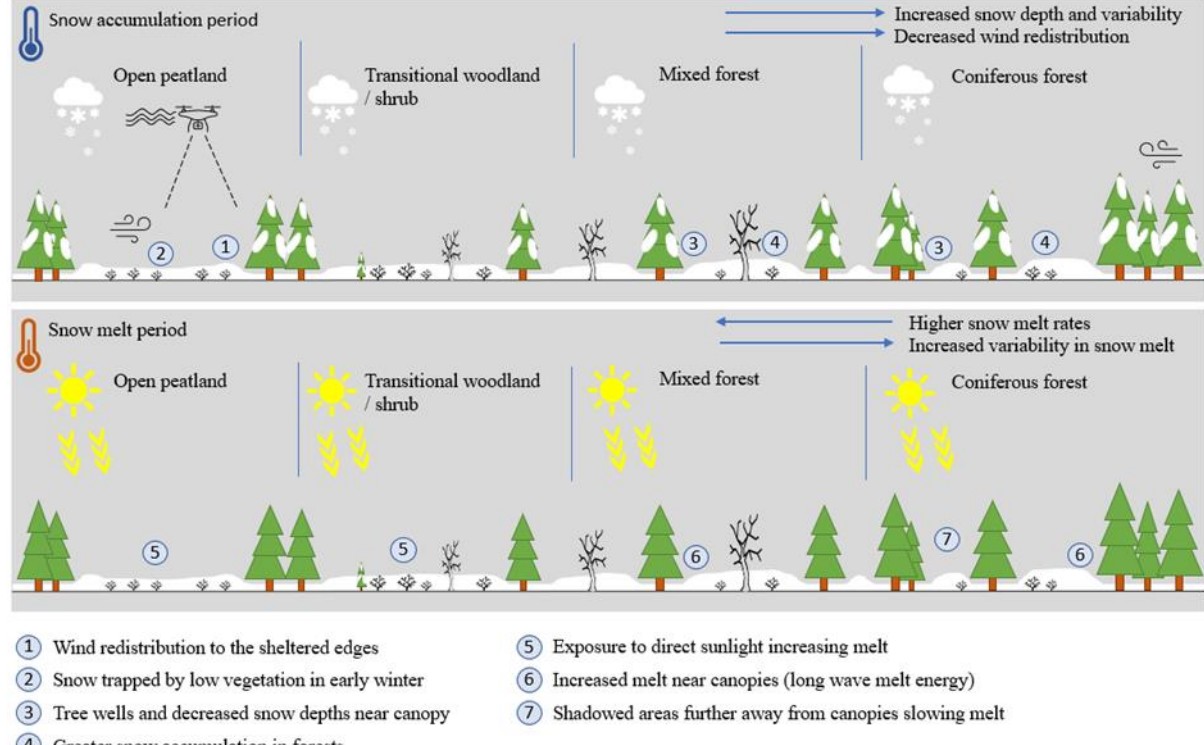

**Fig. 9. Observed snow accumulation and melt processes for different land cover types using the UAS-SfM method.**

## 5.4 Opportunities and challenges in determining snow processes using UAS-SfM

Our results highlight how the UAS-SfM method can be used for frequent, high spatial resolution snow depth coverage in a
cost-efficient manner. The key advantage is that the method allows measurement of snow depth with high position accuracy
(at a centimeter scale) throughout the landscape, which in our 0.5 m resolution maps resulted in 917045 approximations of
snow depth for 23 ha area. This is at least an order of magnitude higher than other established methods, such as snow surveys
with automated magnaprobe (1000–100000 measurement points, Sturm and Holmgren 2018), snow surveys with manual
probes (10–100 measurements, Lundberg and Koivusalo 2003; Pirazzini et al., 2018) or continuous point measurements (1–
10 measurement points, Zhang et al., 2017). The position accuracy of these other established methods is typical +/- 3 m, which
is done using standard GNSS (Global Navigation Satellite System).

A large number of points with high position accuracy allows detailed snow process analysis in relatively large areas or even
up to small catchments using fixed-wing UAVs, and within and between different land cover types that are not easily possible
with other methods, such as manual snow course measurements. Airborne LiDAR has been used for similar analysis with the
advantage of canopy penetration but with a cost which is an order of magnitude higher (Harder et al., 2020). By removing the
parts containing forest canopies with a small buffer, the UAS-SfM method allows analysis of vegetation-snow interactions for



forested areas and larger trees. The observed snow depth peak, especially in forested areas during melt, requires more study and observation.

The greatest challenges in using the UAS-SfM method are related to vegetation, weather, and the reflectance properties of fresh snow. To minimize the vegetation effect, it is recommended to do a bare ground survey soon after snowmelt, when the vegetation is still compact and the growing season has not started. An airborne LiDAR survey of the ground would again allow penetration of the vegetation at a high cost, but we found (Rauhala et al., 2022, submitted) that the difference was minor when tree masks were used to remove larger vegetation. For forested areas, cameras with near-infrared (NIR) frequency bands could help in tree mask creation, using supervised classification, especially for broad-leaved trees whose branches are sometimes mixed with shadows and debris on snow cover. During the mid-winter, the limited daylight hours in high latitudes must also to be considered as they cause good windows for measurements to be rare. The NIR band could further improve snow pixel identification in challenging illumination conditions by avoiding holes in point cloud caused by missing key points (Adams et al., 2018).

The high-resolution snow depth maps generated using the UAS-SfM method could further be used in small-scale (below 1 m to 100 m) studies of snow accumulation and melt processes, including enhanced observation of interactions between snow, vegetation, and topography. On a medium or local scale (100–4000 m), the method could be used to improve landscape specific information for snow depth for recreational use and tourism, and in calibrating/validating catchment scale hydrological models used in research, environmental planning, hydropower, or flood prediction (Kinar and Pomeroy 2015; Sturm 2015; Ala-aho et al., 2017; Hewer and Gough 2018).

## 6 Conclusions

In recent years, UAS-SfM techniques have enabled cost-efficient and high spatial resolution monitoring of snow depth in a variety of land and snow covers. This study extends the coverage of UAS-SfM studies to the subarctic region with multiple surveys through the snow accumulation and melt season (a total of 5 measurement campaigns in three areas) in different weather and illumination conditions. We captured the differences in snow depth variability for subarctic forest and peatland covers, and the increase in this variability as the snow season progresses, especially during snowmelt. Moreover, we successfully generated and used a tree mask to remove trees and the areas immediately next to trees, which are challenging for snow remote sensing, from UAS-SfM derived surface models. We identified multiple theoretically known snow processes and interactions between snow and vegetation, such as canopy interception and wind transport with deposition of snow at forest edges, for forested and peatland areas. The effect of decreased snow accumulation below canopies extending outside the immediate canopy was also shown in a high resolution, spatially extensive analysis. Our study highlights the potential of the UAS-SfM to be used for a detailed study of snow depth in multiple land cover types and snow-vegetation interactions. The data can be used to extend the spatial scale of snow course measurements, in snow model calibration and validation on a catchment scale, and improved forecasts for operational and decision-making purposes.



**Data availability**

The data underlying this analysis and its documentation is available at https://doi.org/10.23729/43d37797-e8cf-4190-80f1-ff567ec62836 (Rauhala et al. 2022) under a Creative Commons CC-BY-4.0 license.

**Author contribution**

LJM, HM, PA and AR designed the field studies, while LJM, AR, AK, and PK carried them out and processed the data. LJM analyzed the data and prepared the manuscript with contributions from all co-authors. HM and PA supervised the research.

**Competing interests**

The authors declare that they have not conflict of interest.

**Acknowledgements**

This study was supported by the Maa- ja vesitekniikan tuki ry, K. H. Renlund Foundation, Academy of Finland (projects 316349, 330319, and ArcI Profi 4), the Strategic Research Council (SRC) decision no. 312636 (IBC-Carbon), EU Horizon 440 2020 Research and Innovation Programme Grant agreement no. 869471, and Kvantum institute at the University of Oulu. We thank Valtteri Höyky and Metsähallitus for assisting with field sampling campaigns. We gratefully acknowledge the field work assistance of Filip Muhic, Kashif Noor, Aleksi Ritakallio, Alexandre Pepy, Jari-Pekka Nousu and Valtteri Hyöky.

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
