# Peer review of "Measuring the spatiotemporal variability of snow depth in subarctic environments using unmanned aircraft systems (UAS) – Part 2: Snow processes and snow-canopy interactions"

_The Cryosphere, 2022_

## Author Comment (AC1)

**General comments**

Authors analysed the spatial and temporal variability of snow depth and interactions between snow and vegetation in a subarctic landscape with coniferous forest, mixed forest, and peatland areas. To determine the variability of snow depth, they used high-resolution maps acquired from four UAS surveys in a one winter season verified by manual snow course measurements and one automatic snow depth station. Authors used interesting approach of creating a tree-mask to remove canopy areas from analysis due to poor penetration of the UAS camera. Authors found that both snow depth and its variability increased with the canopy density. Authors also described the snow depth increase and then decline with a distance from canopy, as well as the increase of the peak value distance from tree as the season progressed.

In my opinion, authors did an interesting work which certainly has a scientific relevance. I think this is an important pure and thorough field study. Therefore, the study has clear potential to be published. However, I do not see much novelty in the study both in terms improving our knowledge or methodological approaches. I see some methodological novelty in using the tree mask to limit the data, but the question is whether it is really novel. I am not saying that the study lacks novelty at all, but I think the authors should better define what is new in the study and how it goes beyond the previous studies. Besides, I have a few other comments listed below, which should be addressed before I can recommend the manuscript for publication.

Response: We thank the reviewer for her/his comments and suggestions. We have now improved the description of novelty and done suggested changes. In below, please find our point-by-point replies to comments and suggestions. Following changes (in italic) were made in introduction to better define the novelty of this study.

L80–L85: "To our knowledge, no studies have used the high-resolution data from UAS-SfM to study snow accumulation and ablation processes, and interactions between vegetation and snow in the subarctic boreal region, *consisting for mosaic of forested and peatland areas with challenging climate factors, such as variable light conditions and very cold temperatures*."

L86–L94: "The overall aim of this study was to evaluate the variability of snow accumulation and melt in high spatial resolution using UAS-SfM. With these novel datasets, we studied interactions between snow cover and vegetation in different subarctic land cover types. *We compared the acquired snow depth data with manual snow transect measurements and assessed the spatial representativeness of a single point snow depth measurement in relation to UAV-SfM derived data. The specific research questions were: 1) how spatiotemporal snow depth variability differ across forested and open mire landscapes, and 2) what canopy controls on this variability can be revealed with high spatial resolution UAS-SfM snow depth surveys."*

**Major comments**

I did not fully understand why authors used CORINE land cover data since they worked with precise UAV based data describing the specific pixel distance from the canopy/trees (which were used to create canopy masks). Maybe I just did not understand it correctly from the text, but why they did not use accurate canopy structure data for the whole analysis? Or was CORINE data used only for general description of the land cover types in individual plots? Please explain it in more detail (probably in methods).

Response: Thank you for pointing out this possibility. We used Corine data because we wanted to have classification method for land cover that is readily available and widely used, and that can be applied for larger areas than are feasible for UAV surveys. The use of Corine data provides a possibility to easily expand the study for larger regions and other areas. The analysis was done with detailed canopy structure data.

Result section 4.3 contains three figures; however, the related text doesn't contain detailed explanation and interpretation (it consists only in three short paragraphs). Please extent the related text substantially to provide the reader with detailed description and interpretation of the related figures.

Response: We will expand the discussion in interpretation of the figures.

In my opinion, conclusion section is too brief and general. I would suggest including more details (including numbers) about individual conclusions. As it is now, it looks like a summary describing what authors did rather than main study conclusions.

Response: Conclusion part will be changed as the reviewer suggested.

**Specific comments**

L 21: One of the study conclusion is that differences between UAS and ground measurements highlights "the poor representation of point measurements even on the sub-catchment scale". This might be certainly true; however, this may also show that point measurement location is not fully representative for the wider area. Could you please add some more discussion related to this issue?

Response: Sentence changed in abstract. L21–L22: "This highlights the poor representation of point measurements in selected location even on the sub-catchment scale."

We also added a sentence in discussion L360–361: "The representativeness of a point measurement location must be considered carefully, not only for a sub-catchment scale but also for wider areas in operational or scientific use."

L 29–31: Authors stated that "This study highlights the potential of the UAS-SfM in high-resolution monitoring of snow depth in multiple land cover types …". UAS is nowadays standardly applied and well-established method for snow depth monitoring (even in diverse vegetations). Therefore, the statement that "it has a potential" might be relevant perhaps 5–7 years ago, but not nowadays. Please consider reformulation.

Response: Potential reformulated to applicability.

L 44-45: Although I agree that individual factors control snow depth at different spatial scales, I do not think that such distinct limit (100 m) can be defined. Maybe consider reformulation.

Response: Added sentence. L47: "However, in nature the distinct limits for factors controlling snow depth at different spatial scales are varying. "

L 79–87: Here authors explain the novelty of their study. Besides others, authors see the novelty in applying UAS imaging in boreal regions. Why this is specifically novel? How the UAS imaging in boreal regions differs from imaging in other areas? I think that application of UAS in boreal regions just because it was never used there before, doesn't mean novelty per se. Please consider more specific explanation.

Response: Reformulated. L82–L85: "To our knowledge, no studies have used the high-resolution data from UAS-SfM to study snow accumulation and ablation processes, and interactions between vegetation and snow in the subarctic boreal region, consisting of a mosaic of forested and peatland areas with challenging climate factors, such as variable light conditions and very cold temperatures."

L 85: One of the research questions is how UAS can be used for snow depth imaging during poor light conditions (probably because the study plots are far beyond the artic cycle). Authors addressed this question rather marginally in Section 5.4, but maybe this might be one of the novel issues which might deserve more attention (see also my general comment and the previous comment).

Response: Thank you for this comment. Similar criticism was found in reviewer comments for manuscript part 1. So, is it possible to expand the discussion in part one?

Fig. 2: Lines with min/max snow depth means snow depth evolution of the one winter season with highest/lowest snow depth or each date on x-axis means maximum/minimum value for this date from all winters at the study period? Please clarify.

Response: Sentence added to Fig. 2 caption: "In x-axis, the min and max snow depth evolution shows the minimum and maximum value for each date from all winters in the period of 2006–2018."

L 123–124: Could you be a bit more specific why two of the surveys were discarded?

Response: Information added, and the sentence was split in three. L132–L134: "Two of the surveys were discarded due to challenges that hindered the data collection. Camera mechanics were frozen in January with very cold temperatures (-30 °C) that caused unfocused pictures. On May, only very small patches of snow, insufficient for the analysis, were remaining in study plots."

Table 1 (and maybe also Table 2 and 3): Consider adding also absolute values of snow depth and not only differences between point measurements and UAS data.

Response: Absolute snow depth added to Table 1. Tables 2 and 3 contains already lots of information and we do not see need to add absolute snow depths to tables 2 and 3.

Fig. 6: What is the physical explanation of increasing differences of the snow depth near canopy with progressing season? How important is the longwave radiation emitted by trees which increases the snowmelt rates near tree trunks? Please discuss shortly.

Response: Discussion added. L382–385: "This variability could be explained by increased shortwave radiation towards spring absorbed by the canopies, thus increasing the emitted longwave radiation that can increase the snowmelt rates near tree trunks. The longwave radiation is a function of tree temperature, which may be significantly different from air temperature and increase as spring progresses (Webster et al 2016)."

L 338: While I generally agree with provided explanation of highest snow depth in forested areas, do you have any data to support this interpretation?

Response: Added sentence. L355–L357: "Even though we do not have direct measurements of interception, our snow survey transect monitoring shows that the snow depths are typically higher in forested landscapes in different years." See data below.

[Figure]

L 345: While this is rather trivial conclusion, I think it might be beneficial for end users (e.g., operational services) and thus it may appear also in the conclusion section.

Response: Added a sentence to the conclusions. L452–L455: "While we found that the widely used snow course data produced a realistic picture of areal snow depth conditions that can be used in operational services, the UAV-SfM derived data can be used to extend the spatial scale of snow course measurements, in snow model calibration and validation on a catchment scale, and improved forecasts for operational and decision-making purposes."

L 357–361: I see the point, however, why it should be interesting? Could you explain it in more detail?

Response: The interesting point is that with the UAV-SfM method this kind of behaviour related probably to the tree well effect and emitted long wave radiation from the canopies could be detected. Sentence was slightly changed.

L378–L379: "Interestingly, with UAV-SfM derived data we detected that the median snow depth had a peak value around 1 m from the tree mask during accumulation season, but this peak distance increased up to 2.5 m in the middle of the melt period."

We also added possible physical explanation for this behaviour:

L382–L385: "This variability could be explained by increased shortwave radiation towards spring absorbed by the canopies, thus increasing the emitted longwave radiation that can increase the snowmelt rates near tree trunks. The longwave radiation is a function of tree temperature, which may be significantly different from air temperature and increase as spring progresses (Webster et al., 2016)."

**Technical corrections**

L 65: Instead of "submitted to the same journal", I would specify its name.

Response: Changed.

Fig. 2: Please consider change of individual line colours/types to increase readability.

Response: Will be done as suggested.

L 362: "For open peatland landcover, this peak may be explained …" It is not clear what "this" refers to. Please consider reformulation.

Response: Details added in sentence:

L387–L388: "For open peatland landcover, this snow depth peak near canopies may be explained by the wind distribution process that transports the snow to the edges of the open areas, where it's trapped by trees."

---

## Author Response (AR1)

**Author's response**

Point by point responses to the referee comments are given below in blue font.

**Comments from the editor:**

Regarding your question about the discussion of poor light conditions, I agree that it would make sense to expand the discussion in Part 1 and then reference that discussion in Part 2.

We thank the editor for the comments. Reference to discussion in Part 2 is added.

Please make sure you are using consistent terminology for in situ snow courses in Part 1 and Part 2. Are "snowlines", "snow transects" and "snow courses" all referring to the same thing? I would suggest using "snow course" and providing a brief description of what that is.

Thank you for pointing this out. We changed the terminology to snow course consistently in Part 1 and 2. The different terms were referring to the same thing. We also slightly modified the description of snow course in Part 2 and made a reference to Part 1 where the term snow course in context of our manuscripts is described in detail.

Manuscript part 2, L145–L146: "During every UAS survey, reference snow depth was manually measured from a standardised snow course (46 stationary points with a mean distance of 50 m between points) transecting the study plots, with an accuracy of +/- 2 cm (see Rauhala et al., 2022 for details)."

**Referee 1:**

**General comments**

Authors analysed the spatial and temporal variability of snow depth and interactions between snow and vegetation in a subarctic landscape with coniferous forest, mixed forest, and peatland areas. To determine the variability of snow depth, they used high-resolution maps acquired from four UAS surveys in a one winter season verified by manual snow course measurements and one automatic snow depth station. Authors used interesting approach of creating a tree-mask to remove canopy areas from analysis due to poor penetration of the UAS camera. Authors found that both snow depth and its variability increased with the canopy density. Authors also described the snow depth increase and then decline with a distance from canopy, as well as the increase of the peak value distance from tree as the season progressed.

In my opinion, authors did an interesting work which certainly has a scientific relevance. I think this is an important pure and thorough field study. Therefore, the study has clear potential to be published. However, I do not see much novelty in the study both in terms improving our knowledge or methodological approaches. I see some methodological novelty in using the tree mask to limit the data, but the question is whether it is really novel. I am not saying that the study lacks novelty at all, but I think the authors should better define what is new in the study and how it goes beyond the previous studies. Besides, I have a few other comments listed below, which should be addressed before I can recommend the manuscript for publication.

Response: We thank the reviewer for the helpful comments and suggestions. We have now improved the description of novelty and done suggested changes. In below, please find our point-by-point

replies to comments and suggestions. Following changes were made in introduction to better define the novelty of this study.

L79–L82: "The previous studies have shown technical advancements and improved accuracies of UAS-SfM in snow monitoring. However, to our knowledge, no studies extend the focus on the snow processes: accumulation and ablation, and interactions between vegetation and snow in the subarctic boreal region, consisting of mosaic of forested and peatland areas with challenging climate factors, such as variable light conditions and very cold temperatures."

L83–L89: "The overall aim of this study was to evaluate the variability of snow accumulation and melt in high spatial resolution using UAS-SfM. With these novel datasets, we studied interactions between snow cover and vegetation in different subarctic land cover types. We compared the acquired snow depth data with manual snow course measurements and assessed the spatial representativeness of a single point snow depth measurement in relation to UAV-SfM derived data. The specific research questions were: 1) how spatiotemporal snow depth variability differ across forested and open mire landscapes, and 2) can we attribute the landscape differences to snow-canopy interaction processes using high spatial resolution UAS-SfM snow depth surveys."

**Major comments**

I did not fully understand why authors used CORINE land cover data since they worked with precise UAV based data describing the specific pixel distance from the canopy/trees (which were used to create canopy masks). Maybe I just did not understand it correctly from the text, but why they did not use accurate canopy structure data for the whole analysis? Or was CORINE data used only for general description of the land cover types in individual plots? Please explain it in more detail (probably in methods).

Response: Thank you for pointing out this possibility. We used Corine data because we wanted to have classification method for land cover that is readily available and widely used, and that can be applied for larger areas than are feasible for UAV surveys. The use of Corine data provides a possibility to easily expand the study for larger regions and other areas. The analysis was done with detailed canopy structure data.

Following changes were made in manuscript L184–L186: "The Corine land cover dataset (EEA, 2018) was selected for this purpose due to its i) harmonized vegetation classification and ii) availability across the Eurasian region, that readily enables expanding the studies for larger regions and other areas."

Result section 4.3 contains three figures; however, the related text doesn't contain detailed explanation and interpretation (it consists only in three short paragraphs). Please extent the related text substantially to provide the reader with detailed description and interpretation of the related figures.

Response: The discussion is extended to explain and interpret the figures (L269–290).

In my opinion, conclusion section is too brief and general. I would suggest including more details (including numbers) about individual conclusions. As it is now, it looks like a summary describing what authors did rather than main study conclusions.

Response: Conclusion section substantially modified.

**Specific comments**

L 21: One of the study conclusion is that differences between UAS and ground measurements highlights "the poor representation of point measurements even on the sub-catchment scale". This might be certainly true; however, this may also show that point measurement location is not fully representative for the wider area. Could you please add some more discussion related to this issue?

Response: Sentence changed in abstract. L21–L22: "This highlights the poor representation of point measurements in selected location even on the sub-catchment scale."

We also added a sentence in discussion L367–368: "The representativeness of a point measurement location must be considered carefully, not only for a sub-catchment scale but also for wider areas in operational or scientific use."

L 29–31: Authors stated that "This study highlights the potential of the UAS-SfM in high-resolution monitoring of snow depth in multiple land cover types …". UAS is nowadays standardly applied and well-established method for snow depth monitoring (even in diverse vegetations). Therefore, the statement that "it has a potential" might be relevant perhaps 5–7 years ago, but not nowadays. Please consider reformulation.

Response: Potential reformulated to applicability.

L 44-45: Although I agree that individual factors control snow depth at different spatial scales, I do not think that such distinct limit (100 m) can be defined. Maybe consider reformulation.

Response: Added sentence, L46: "However, in nature the distinct limits for factors controlling snow depth at different spatial scales are varying. "

L 79–87: Here authors explain the novelty of their study. Besides others, authors see the novelty in applying UAS imaging in boreal regions. Why this is specifically novel? How the UAS imaging in boreal regions differs from imaging in other areas? I think that application of UAS in boreal regions just because it was never used there before, doesn't mean novelty per se. Please consider more specific explanation.

Response: Reformulated. L80–L82: "However, to our knowledge, no studies extend the focus on the snow processes: accumulation and ablation, and interactions between vegetation and snow in the subarctic boreal region, consisting of mosaic of forested and peatland areas with challenging climate factors, such as variable light conditions and very cold temperatures."

L 85: One of the research questions is how UAS can be used for snow depth imaging during poor light conditions (probably because the study plots are far beyond the artic cycle). Authors addressed this question rather marginally in Section 5.4, but maybe this might be one of the novel issues which might deserve more attention (see also my general comment and the previous comment).

Response: Thank you for this comment. The poor light conditions are certainly one of the novel issues, especially in comparison of different equipment. As also editor agreed, the discussion about light conditions was expanded in Part1 and referred in Part2.

In addition to changes in introduction we added sentences to manuscript section 5.4, L441–L442: "The light conditions and extreme cold weather pose a particular challenge in Northern boreal zone. The issues related to this dataset are addressed in detail by (Rauhala et al 2022)."

Fig. 2: Lines with min/max snow depth means snow depth evolution of the one winter season with highest/lowest snow depth or each date on x-axis means maximum/minimum value for this date from all winters at the study period? Please clarify.

Response: Sentence added to Fig. 2 caption: "The min and max snow depth shows the minimum and maximum value for each date from all winters in the period of 2006–2018."

L 123–124: Could you be a bit more specific why two of the surveys were discarded?

Response: Information added, and the sentence was split in three. L126–L129: "Two of the surveys were discarded due to challenges that hindered the data collection. Camera mechanics froze due to very cold temperatures during the January survey, causing unfocused pictures. In May, only very small patches of snow, insufficient for the analysis, were remaining in study plots."

Table 1 (and maybe also Table 2 and 3): Consider adding also absolute values of snow depth and not only differences between point measurements and UAS data.

Response: Tables were changed substantially. Absolute snow depths were added to Tables 1, 2 and 3, with difference compared to point measurement. Point measurement data was also added to Tables 1 and 2.

Fig. 6: What is the physical explanation of increasing differences of the snow depth near canopy with progressing season? How important is the longwave radiation emitted by trees which increases the snowmelt rates near tree trunks? Please discuss shortly.

Response: Discussion added. L389–394: "This variability could be explained by increased shortwave radiation towards spring absorbed by the canopies, thus increasing the emitted longwave radiation that can increase the snowmelt rates near tree trunks. The longwave radiation is a function of tree temperature, which may be significantly different from air temperature and increase as spring progresses (Webster et al 2016). Because the differences increase specifically during the melt period, we attribute the increase to the tree longwave radiation. During the snow accumulation period, we propose the canopy interception to be the main driver in spatial snow depth variability.

L 338: While I generally agree with provided explanation of highest snow depth in forested areas, do you have any data to support this interpretation?

Response: Added sentence. L363–L364: "Even though we do not have direct measurements of interception, our snow course survey monitoring shows that the snow depths are typically higher in forested landscapes in different years." See data below.

[Figure]

L 345: While this is rather trivial conclusion, I think it might be beneficial for end users (e.g., operational services) and thus it may appear also in the conclusion section.

Response: Added a sentence to the conclusions. L467–L470: "While we found that the widely used snow course data produced a realistic picture of areal snow depth conditions that can be used in operational services, the UAV-SfM derived data can be used to extend the spatial scale of snow course measurements, in snow model calibration and validation on a catchment scale, and improved forecasts for operational and decision-making purposes."

L 357–361: I see the point, however, why it should be interesting? Could you explain it in more detail?

Response: The interesting point is that with the UAV-SfM method this kind of behaviour related probably to the tree well effect and emitted long wave radiation from the canopies could be detected. Sentence was slightly changed.

L385–L386: "Interestingly, with UAV-SfM derived data we detected that the median snow depth had a peak value around 1 m from the tree mask during accumulation season, but this peak distance increased up to 2.5 m in the middle of the melt period."

We also added possible physical explanation for this behaviour:

L389–L394: "This variability could be explained by increased shortwave radiation towards spring absorbed by the canopies, thus increasing the emitted longwave radiation that can increase the snowmelt rates near tree trunks. The longwave radiation is a function of tree temperature, which may be significantly different from air temperature and increase as spring progresses (Webster et al., 2016). Because the differences increase specifically during the melt period, we attribute the increase

to the tree longwave radiation. During the snow accumulation period, we propose the canopy interception to be the main driver in spatial snow depth variability."

**Technical corrections**

L 65: Instead of "submitted to the same journal", I would specify its name.

Response: Changed.

Fig. 2: Please consider change of individual line colours/types to increase readability.

Response: Done as suggested.

L 362: "For open peatland landcover, this peak may be explained …" It is not clear what "this" refers to. Please consider reformulation.

Response: Details added in sentence:

L395–L396: "For open peatland landcover, this snow depth peak near canopies may be explained by the wind distribution process that transports the snow to the edges of the open areas, where it is trapped by trees."

**Referee 2:**

**General Comments**

The authors present a comparison of snow depths observed using UAV-based structure from motion. They explore the relationship of snow depth through a snow accumulation and melt season across various land cover types within a sub-arctic environment, paying special attention to the interactions between forest canopies and snow depth. The main comparison techniques include presenting differences between UAV-SfM observations and a single observing station, between land cover classes, and based on the distance from the canopy. I believe the time-series component mixed with both the spatial coverage and diversity of land cover types makes this work a valuable addition to the literature. I also agree with Reviewer 1 in that the tree masking was an interesting component and additionally think that the exploration into snow depth as a function of canopy distance was particularly informative. The use of a summary figure at the end of the manuscript was also valuable.

With that being said, I do believe there is also substantial room for improvement. The objectives stated in the introduction did not seem to align with the presented results. Also, while the dataset is impressive in its own right, the analysis and statistical methods were unclear or questionable at times. I have included some specific suggestions outlining where improvements can be made within the following sections. I think the paper is novel enough to warrant publishing with some revision.

Response: Thank you for the encouraging comments. Objectives were aligned more carefully with the results and the analysis, and the statistical methods were clarified. Detailed point to point responses are added below.

**Major Suggestions/Comments**

L84–87: These objectives don't seem to line up with what was addressed in the study. The first point seems to be the focus of the accompanying paper (not this one). The second and third points should be kept. Though, consider adjusting them to be more in line with the actual analyses done (1 – comparing spatiotemporal snow depth variability across different land cover types, and 2 – exploring

the controls canopy has on this variability). There is also a large component in the way the results are presented that presents all observations relative to the point observation. Since assessing the spatial representativeness of the single-point site is such a focus of the analysis approach, I suggest adding a clear mention of this within the objectives as well.

Response: Thank you for pointing out this mismatch on the objectives. We reformulated the objectives accordingly and added sentence about single point measurement assessment.

L83–L89: "The overall aim of this study was to evaluate the variability of snow accumulation and melt in high spatial resolution using UAS-SfM. With these novel datasets, we studied interactions between snow cover and vegetation in different subarctic land cover types. We compared the acquired snow depth data with manual snow course measurements and assessed the spatial representativeness of a single point snow depth measurement in relation to UAV-SfM derived data. The specific research questions were: 1) how spatiotemporal snow depth variability differ across forested and open mire landscapes, and 2) can we attribute the landscape differences to snow-canopy interaction processes using high spatial resolution UAS-SfM snow depth surveys."

The word 'variability' is used throughout when referencing the difference between the 5th and 95th percentiles of the snow depth distribution. This serves more as a measure of the range, not variability (like standard deviation/variance). Please revise your use of the word 'variability' throughout (or update the statistical method to better reflect variability). In most cases, it could be replaced with the word range.

Response: Done as suggested.

L230–244: This section and potentially the statistical approach should be restructured. Initially you mention that all classes are significantly different with high confidence (very low p-value), then process to counter this claim when using the smaller random samples of snow depth data. What is the takeaway here? I suggest selecting a single appropriate statistical test and sticking with it.

Response: Thank you for the good comment. In our opinion it's important to show the complete process for the reader because in using UAS-SfM method the amount of datapoints is usually very large and others could run in similar challenges with sample sizes that can be too large to fail. In the new version of the manuscript, we explain in a more detailed way why our original approach serves study better.

Clarified in results. L246–L248: "However, using the UAS-SfM method, the amount of datapoints is very large, potentially making it difficult for the test to accept the null hypothesis (Lin et al., 2013). To address this, we reduced the sample size with random sampling to highlight the true differences between the land cover types."

With such a considerable focus on comparisons between the ultrasonic sensor at Kenttarova and the UAV-SfM observations, there needs to be a more comprehensive discussion as to the land cover surrounding this site (i.e., distance from canopy, understory, Corine class canopy type etc.).

Response: Details of the location added. L149–150: "The Corine classification of Kenttärova snow depth sensor location is coniferous forest, the distance to the canopies is approximately 5 m and the understory in the sensor location is replaced with artificial green grass mat."

Can the vegetation classification (using Corine) be enhanced by using ortho mosaic data & your tree masks? As is, the resolution is somewhat limiting, and it is difficult to tell how effectively this

captures the different canopy types. There would be considerable value in adding forest type & density information into the analysis, without adding much additional work.

Response: We used Corine dataset since it provides widely used and standardised method for canopy type. Corine is also available for larger regions and thus allows further cross-checking with future studies. We see potential for more high-resolution canopy cover information gained from UAVs but this would be already another scientific study/manuscript.

Sentence added/modified in manuscript L184–L186: "The Corine land cover dataset (EEA, 2018) was selected for this purpose due to its i) harmonized vegetation classification and ii) availability across the Eurasian region, that readily enable expanding the future studies for larger regions and other areas."

**Minor/Technical Suggestions**

L24: First instance of using variability in place of range. Please adjust the terminology here (and throughout)

Response: Done.

L30–31: This point should be modified to reflect the fact that even if there is field data (collected in a classic way through point sites/snow courses) snow analyses are still limited. Doing so would make this more in line with the points made in the discussion and conclusions later in the paper.

Response: Done, sentence added. L29–L31: "This study highlights the applicability of the UAS-SfM in high-resolution monitoring of snow depth in multiple land cover types and snow-vegetation interactions in subarctic and remote areas where field data is not available, or where the available data is collected using classic point measurements or snow courses."

L49–51: "In forests,…." This sentence is a bit challenging to read, please consider revising

Response: Revised and sentence is split in two, manuscript L48–L50: "In forests, the snow depth has been found to depend on canopy cover with less snow in the denser forest because of canopy interception and sublimation (Varhola et al., 2010). In forest openings, the snow depth depends, among other things, on their size, with most snow accumulating in clearings 2–5 times the height of nearby trees (Pomeroy et al., 2002)."

L59: remove 'the scale of', is redundant

Response: Done.

L68: Please specify more clearly that a more comprehensive review of UAV-SfM studies is included in the accompanying paper

Response: Done, manuscript L68–L69: "A more comprehensive review of UAV-SfM studies is included in the accompanying paper, Rauhala et al. (2022)."

L71: For the likely audience of this work, an explicit definition of tree wells is unnecessary

Response: Definition removed.

L80: Just because the region is locationally different doesn't necessarily make this work different. Please mention some of the unique considerations (like lighting, forest structure, snow properties) that make the subarctic region a unique study area.

Response: Done. Added sentence. L84–L85: "…consisting for mosaic of forested and peatland areas with challenging climate factors, such as variable light conditions and very cold temperatures."

L113: Please clean this up a bit, the use of parenthesis is excessive and challenging to associate the numbers mentioned with the cited works

Response: Cleaned.

L149: I assume 'high-quality and moderate depth filtering settings' are specific to the software used. Can you please provide a reference to what these parameters mean, or briefly add description herein

Response: These are indeed software specific settings. Reference to Agisoft user manual added to manuscript. The parameters are briefly described below.

The quality parameter is a 5-step (Ultra High, High, Medium, Low, Lowest) setting specifying image downscaling during the depth map generation. Ultra high quality utilizes original photos, while each following step downscales preliminary image size by a factor of 4. The Ultra High quality setting has considerably longer processing times, and more importantly, requires a very large amount of random access memory for large datasets.

Depth filtering is a 4-step (Disabled, Mild, Moderate, Aggressive) setting specifying the aggressiveness of outlier filtering during the dense cloud generation. Aggressive filtering is generally recommended for aerial data processing but Mild filtering might be more suitable for example in the case of poorly textured roofs, etc. Our testing with the December 2018 dataset indicated that Moderate depth filtering produced all around best results.

Reference:

Agisoft Metashape User Manual: Professional Edition, Version 2.0. Agisoft LLC. Available online: https://www.agisoft.com/pdf/metashape-pro_2_0_en.pdf, 2023.

L165: You introduce DoDs here, then quickly shift to referring to them as snow depth maps. Am I correct that they are the same? If so (or if not) make sure it is clear that they are interchangeable terms (or not).

Response: Yes, they are interchangeable here. Short sentence added to manuscript L172–273: "DoD is used here interchangeably with snow depth map."

L178: What does 'i) approved vegetation classification' mean in this context?

Response: We added reference for European Union level Corine datasets and its classification and changed "approved" to "harmonized".

L199: Please re-iterate that the 'point site' refers to the single automated station located within the forest

Response: Done.

Table 1: Please clearly indicate which depth is subtracted from which in the figure caption. Also, while presenting only the differences between depths at the courses and to the observing site is interesting (and relevant to the spatial representativeness question), I think this table would greatly benefit from the inclusion of the actual median depth values by class. For example, include the median depth followed by the difference relative to the point site (i.e., 55 cm (-10 cm))

Response: Actual median values added. Table caption changed to: "Median snow depths of the UAS-SfM derived DoDs for subplots, whole survey area (All) and snow course. Differences between the snow depth median (cm) and the point measurement (Point) are presented in brackets."

L216: Please clarify 'point measurements' – is this plural to represent the time-series at the single observing site, or does it refer to the snow course obs. Please try to make this clear throughout the paper.

Response: Refers to the single point measurements. Corrected throughout the manuscript.

Table 1 -> Section 4.2 (and other locations): Please make sure your use of units is consistent (pick m or cm)

Response: Done throughout the manuscript.

L219–L223: I was confused by the statistics here. Why not just present the median depths?

Response: We improved this section to be crisper. Presenting the ranges in addition the median snow depths gives more information about the landcover effect on snow depth variability.

L224: "difference in variability" (?)

Response: Corrected

L225: 'difference' compared to what? The point reference?

Response: Revised. Compared to the single point reference.

Table 2-3: Again, in my opinion, there is an unnecessary added layer of complexity here. I suggest presenting the true depth values (5-95%) within each field and timestep, then presenting the observations for the same timestep at the point site & the range/median of the snow courses. One idea… a time-series figure (or 1 per plot) with the data bounds may be a good additional way to visualize changes occurring across each class (& their relationship with the observations). -> it also would capture similar information to what is shown in Figure 5

Response: Tables 2 and 3 were changed as suggested. True depth values are used and additionally true depth values of point site were added to table 2.

Figure 5: This is a great figure, please ensure it is referenced/discussed sufficiently in the text

Response: Discussion and one ref added. L238–L240: "The increase in snow depth variability can be clearly observed in Fig. 5 that shows the snow depth distributions and median snow depths for each survey time and landcover type. The distributions generally follow normal distribution with increasing tail lengths towards the end of winter."

Line 263: Again, variability needs to be clearly defined

Response: Clarified.

Figure 6-8: Since these plots are showing similar information, I think their number can be reduced. It is not clear to me what Figure 8 adds to the paper. Consider revising.

Response: We would prefer to keep the Figure 8. It shows that the number of data points gets scarce further away from the canopy. This figure shows that the statistical uncertainty gets substantially higher with less data points. We extended chapter 4.3 to further discuss and interpret the figures.

Section 5.1: This section is well written but lacking a bit in terms of the actual findings regarding spatiotemporal variability during the accumulation and melt season. Consider adding to the discussion using relevant statistics from the paper relevant to the spatiotemporal variability

Response: Thank you for the good suggestions. We added relevant statistics to section 5.1. More details for seasonal and land cover specific snow depth variability are included in section 5.2.

Manuscript modifications in section 5.1. L308–315: "As the generated snow depth maps were compared favorably to conventional snow course surveys (difference in median snow depth between snow course and UAS-SfM based data for the whole survey area being 4 cm in early snow accumulation and 1 cm in the melt season), the high-resolution method extends areal coverage, providing more detailed information of snow depth distribution and data for the snow process analysis. Moreover, the spatial variability shows the errors that may be associated with using point measurement as a regional reference for snow depth (Figs. 4 and 5). Regional snow depth/presence would be greatly overestimated (22 cm for whole survey area, varying from 28 cm in open peatland to 8 cm in forest) during the melt season if point measurement in the forest was used for regional reference."

L308, 310: careful using 'variability' here

Response: Changed to range.

L313–314: sentence here is a bit wordy

Response: Revised. L336–L338: "The low vegetation in open areas can hinder wind transport close to the ground (Liston et al., 2002), but its effect will diminish when the snow depth reaches its height. After that, the fallen snow is more subject to wind redistribution."

L326: Note the magnitude of the variability/range

Response: Done.

L342: Do you mean 'similar' types here? Or 'different'?

Response: Similar as the ultrasonic station is also located in forest. Small change made and sentence added concerning representativeness of point measurement location. L365–368: "At the end of the snow accumulation period and especially the middle of the melt period, the snow depths were substantially lower than the ultrasonic point measurement for forested and transitional land covers, highlighting the poor representativity of point measurements even for similar land cover types (Fig. 5). The representativeness of a point measurement location must be considered carefully, not only for a sub-catchment scale but also for wider areas in operational or scientific use."

L344–345: Snow courses are good, but I think it would be useful to reiterate that they are limited in their ability to describe the types of canopy interactions observed in this work (10's-100's of obs, vs. thousands)

Response: Sentence added. L370–L372: "Snow course measurements are limited in their ability to describe detailed canopy interactions with their low number of observations (tens to hundreds) compared to the UAS-SfM method (up to millions)."

L365: '….by the limited canopy effect,…'

Response: Corrected.

L380: errant '('

Response: Corrected.

L388: completely up to the authors, but I think a more reasonable range of magnaprobe survey observations is in the 1,000-10,000 range. 100,000 seems a bit high from my experience. This also helps to make the value of the UAV-SfM clearer

Response: Changed to 10000. 100000 might be an overkill.

L411: flip local & medium

Response: Flipped.

L419: consider removal of 'illumination conditions'. While mentioned in the part 1 paper, this does not seem to be something addressed in this manuscript

Response: Removed.